# Increase in short telomeres during the third trimester in human placenta

**Paula K. Edelson**[1]*, **Michala R. Sawyer**[2], **Kathryn J. Gray**[3], **David E. Cantonwine**[3], **Thomas F. McElrath**[3], **Mark Phillippe**[2]

**1** Division of Maternal-Fetal Medicine, Department of Obstetrics and Gynecology, University of Pennsylvania, Philadelphia, Pennsylvania, United States of America, **2** Division of Maternal-Fetal Medicine, Department of Obstetrics and Gynecology, Massachusetts General Hospital, Boston, Massachusetts, United States of America, **3** Division of Maternal-Fetal Medicine, Department of Obstetrics and Gynecology, Brigham and Women's Hospital, Boston, Massachusetts, United States of America

* kaitlyn.edelson@pennmedicine.upenn.edu

**Data Availability Statement:** The data underlying the results presented in the study are available from the Phillippe Lab at the Vincent Center for Reproductive Biology at Massachusetts General

## Abstract

An increase in telomere shortening in gestational tissues has been proposed as a mechanism involved in the timing for the initiation of parturition. An increase in very short telomeres with increasing gestational age has been observed in mice; this study sought to explore this phenomenon in human pregnancies. Specifically, this study addressed the hypothesis that prior to labor, the quantity of very short telomeres (<3 kilobase (kb) lengths) increases in human placental tissue as term gestation approaches. The primary outcome was the quantity of very short telomeres present in placental tissue. Quantitative measurements of very short telomeres were performed using real-time polymerase chain reaction (qPCR) adaptation of the telomere restriction fragment technique. Placental tissue from 69 pregnant individuals were included. Mean gestational age was 39.1 weeks (term) and 36.2 weeks (preterm). For term versus preterm placentas, the observed increase in very short telomeres were as follows: 500 bp telomeres increased by 1.67-fold ($p < 0.03$); 1 kb telomeres increased 1.67-fold ($p < 0.08$); and 3 kb telomeres increased 5.20-fold ($p < 0.001$). This study confirms a significant increase in very short telomeres in human placental tissue at term; thereby supporting the hypothesis that telomere shortening at term contributes to the mechanism that determine the length of pregnancy thereby leading to onset of parturition.

## Introduction

Despite advancements in the understanding of the molecular, endocrine and inflammatory mechanisms occurring during the initiation of human labor (parturition), the biologic clock mechanism(s) that determine the length of gestation leading to the onset of human parturition are not well understood. Telomere length, specifically telomere shortening over time, has been observed to be a major component of the biologic clock that determines life-span in adult humans [1, 2]. Using this premise, we have proposed a telomere gestational clock hypothesis in which shortening of telomeres in placental trophoblasts and fetal membranes (i.e., gestational tissues) leads to cellular apoptosis, resulting in the release of increasing amounts of cell-

Hospital (contact MPHILLIPPE@mgh.harvard.edu).

**Funding:** KJG is supported by NIH/NHLBI K08 HL146963. The funders had no role in study design, data collection and analysis, decision to publish, or preparation of the manuscript. MP is supported by Burroughs-Wellcome Fund. The funders had no role in study design, data collection and analysis, decision to publish, or preparation of the manuscript. There was no additional external funding received for this study.

**Competing interests:** The authors have declared that no competing interests exist.

free DNA that stimulates a pro-inflammatory response [3, 4], ultimately leading to the spontaneous onset of labor as extensively discussed in the recently published review by Phillippe [5].

The average telomere lengths in blood and skin samples among newborn humans range from about 9 to 12.5 kilobases (kb) [6]. Previous work has demonstrated that placental telomere length decreases with increasing gestational age [7]. Prior studies of telomere lengths in human pregnancies have largely focused on measuring average telomere length [7–10], rather than the proportion of "very short" telomeres (VST) (i.e., telomere segments at or less than 3 kb in length). Some studies have also reported telomere shortening as a percentage of total telomere length in gestational tissue in response to maternal obstetric complications, such as gestational diabetes [11]. However, increases in the quantity of short telomeres, and not reductions in the average telomere lengths, are thought to be the effectors of telomere dysfunction, leading to cellular senescence, apoptosis, aging, and tissue dysfunction [12]. The increase of VST during late gestation has, therefore, been proposed as a biologic clock signal leading to apoptosis in the gestational tissues and the stimulation of proinflammatory signaling ultimately leading to labor.

In mouse studies, very short telomere segments in the placenta and fetal membranes have been observed to increase at the end of gestation compared to mid-gestation [13]. To date, only one recently published report has described a significant increase in short telomeres at term in human placental and chorioamniotic membrane tissues compared to 18 weeks of gestation [13]. The purpose of this study was to further test the telomere gestational clock hypothesis in human pregnancy by assessing the relative quantity of VST during the middle of the third trimester of pregnancy (average 36 weeks), as compared to the end of the third trimester (average 39 weeks) (i.e. a time period preceding the spontaneous onset of labor (parturition)).

## Materials and methods

Placental samples were collected immediately following Cesarean delivery at two hospitals within the Mass General Brigham HealthCare System in Boston, MA. Inclusion criteria for samples were as follows: singleton pregnancies between 32 to 41 weeks and delivery by cesarean section prior to the onset of labor. Women in labor were excluded to avoid capturing any potential effect of labor on telomere length. Institutional review board approval was granted by the Partners Human Research Committee, and the need for informed consent was waived and thus not required to be obtained. Clinical and demographic data were collected by medical record review. Those placentas meeting inclusion criteria were collected and samples were taken from four representative 1–1.5 cm full thickness sections of the placental disk with a scalpel. After collection, the placental tissues were immediately rinsed in phosphate-buffered saline and stored at -80˚C until utilized for DNA extraction.

The DNA was extracted from placental tissues using High Pure PCR Template Prep kits (Roche Applied Science) according to the manufacturer's protocol. The concentration and quality (based on the 260/280 nm ratio) of the isolated DNA was determined using a Nano-Drop spectrophotometer (ThermoFisher Scientific). To quantify the increase in very short telomere segments (i.e., 500 base, 1 kb, and 3 kb lengths), a quantitative PCR (qPCR) modification of the telomere restriction fragment (TRF) technique was utilized [14]. Specifically, the DNA underwent restriction endonuclease treatment using *Hinf I* and *Rsa I* to digest the genomic DNA (but not the telomere DNA), thereby releasing intact telomere segments. The treated DNA was then size fractionated using the E-Gel Power Snap Electrophoresis system (ThermoFisher Scientific) with collections of approximate 500 base, 1 kb, and 3 kb DNA fractions. Real-time qPCR was performed in triplicate using the telomere PCR primers reported by Gil and Coetzer [15], the SsoAdvanced Universal SYBR Green Supermix (BioRad), and 5 ng DNA

aliquots. The qPCR cycles were run using the CFX96 Touch Real-Time PCR Detection System as follows: 40 cycles at 95˚C x 10 sec and 55.7˚C x 30 sec. Using the same DNA samples, real-time qPCRs were performed using primers and amplification cycles optimal for the human GAPDH gene, which served as the DNA control.

The Pfaffl Method [16] was used to calculate the increase in the relative quantity of very short telomere DNA at term compared to preterm. Based on prior mouse data [14], power calculations suggested that a sample size of 13 would be needed to detect a two-fold difference in very short telomeres between term and preterm groups. The data were analyzed using the Kruskal-Wallis ANOVA on Ranks and multiple comparisons tests (using the Dunn method) with significance at $p \leq 0.05$.

## Results

Placental tissue was analyzed from 58 term pregnancies ($\geq$37 weeks) and 11 preterm ($<$37 weeks) pregnancies. The mean gestational ages at delivery were 39 weeks 1 days (term) and 36 weeks 2 days (preterm). Maternal and fetal characteristics between the term and preterm pregnancies did not differ significantly (Table 1). The mean maternal age was 33.2 years for the preterm group, and 34.6 years for the term group. Pre-pregnancy Body Mass Index (BMI) was 25.1 in the preterm group and 27.3 in the term group. Hypertension was present among 2/11 (18%) of the preterm group and only 2/58 (3%) of the term group. Smoking was uncommon between both groups, with no smoking in the preterm group and only 2/58 (3%) in the term group. As expected, the average birthweights were lower in the preterm group: 2977 grams for the preterm versus 3545 grams for the term group. Indications for delivery by Cesarean were similar between the two groups, including fetal malpresentation, prior Cesarean delivery, and prior uterine surgery.

The relative quantities of very short telomeres (VST) segments were significantly increased in term as compared to preterm placentas as follows: 500 bp telomere segments = 1.67-fold increase (median, interquartile range 0.86–4.16, $p < 0.03$) and 3 kb telomere segments = 5.20-fold increase (median, interquartile range 3.01–8.41, $p < 0.001$), as shown in Fig 1 and Table 2. There was a similar, but not statistically significant trend in the relative increase in the quantity of 1 kb telomere segments for the term compared to the preterm placental groups (Table 2).

**Table 1. Maternal and fetal characteristics of term and preterm placentas.**

|  | Preterm ($<$37 weeks) | Term ($\geq$37 weeks) |
|---|---|---|
|  | n = 11 | n = 58 |
| Gestational age at delivery | 36 weeks 2 days | 39 weeks 1 days |
| Maternal age (years) | 33.2 | 34.6 |
| Pre-pregnancy BMI (kg/m$^2$) | 25.1 | 27.3 |
| Tobacco use | 0 (0%) | 2 (3%) |
| Hypertension | 2 (18%) | 3 (5%) |
| Race Caucasian | 6 (55%) | 40 (69%) |
| Fetal sex male | 2 (18%) | 29 (50%) |
| Insurance private | 7 (64%) | 30 (52%) |
|  |  | 28 (48%) |
| Indications for delivery | Breech | Prior Cesarean delivery |
|  | Prior Cesarean delivery | Prior myomectomy |
|  |  | Breech |

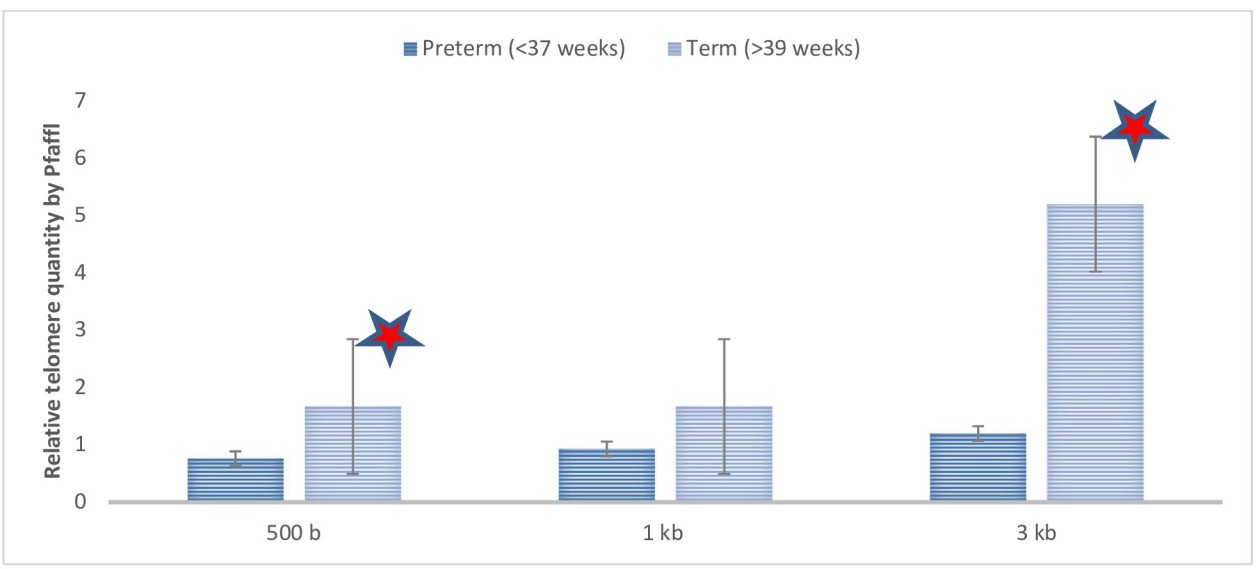

Data in mean ± S.D. (standard deviation), N = 11 for preterm and N = 58 for term samples. (★)

indicates p < 0.05 comparing term to preterm.

**Fig 1. Relative telomere quantity in preterm versus term human placental tissue by telomere lengths.** Relative quantity of very short telomeres (VST), stratified by telomere length, preterm group serves as reference for Pfaffl calculation. Data in mean ± S.D. (standard deviation), N = 11 for preterm and N = 58 for term samples. (·) indicates p < 0.05 comparing term to preterm.

## Discussion

For human placental tissue, we have observed that the relative quantity of 500 bp and 3 kb length telomere segments were significantly higher in term compared to preterm placental tissues. This observation supports the hypothesis that telomere shortening in the placenta occurs with the progression of gestational toward term in our cohorts of placentas obtained from pregnant women delivered pre-labor.

The phenomenon of telomere shortening prior to the onset of labor, with an increase in VST, may help to better understand the pathways that lead to the onset of parturition. Physiologic senescence of the gestational tissues (i.e. the placenta and fetal membranes) occurs as term approaches and is accelerated by the increased oxidative stress of term pregnancy [17]. Progressive aging of the gestational tissues, cellular apoptosis and the release of proinflammatory mediators (including cell-free DNA) have been associated with the onset of parturition

**Table 2. Relative quantity of very short telomeres (VST) in human placental tissue.**

| Telomere fragment size | Preterm (<37 weeks) | Term (≥37 weeks) | *P*-value |
|---|---|---|---|
| | N = 11 | N = 58 | |
| | Median, (IQR) | Median, (IQR) | |
| 500 bp size | 0.76 (0.27–2.08) | 1.67 (0.86–4.16) | 0.03 |
| 1 kb size | 0.93 (0.48–1.47) | 1.67 (0.87–3.48) | 0.08 |
| 3 kb size | 1.2 (0.85–1.45) | 5.2 (3.01–8.41) | <0.001 |

Results presented as median (interquartile range) of Pfaffl relative quantity of telomere length.

[18]. Our studies reported here have shown a parallel increase in short telomeres at term in human placental tissue, thereby supporting the hypothesis that placental senescence in term pregnancy may be physiologic and play a role in determining the timing for the initiation of parturition.

Most prior work on telomeres in human placental tissue examined average telomere lengths and reported an average shortening of telomere length across pregnancy [7, 9, 10]. However, a recent study by Lai et al. [13] has evaluated short telomere segments in human gestational tissues, observing a significant increase of telomere segments less than 3 kb in both term placentas and chorioamniotic membranes when compared to these same tissues obtained at 18 weeks of gestation. Interestingly, these investigators did not observe a corresponding decrease in mean telomere lengths in their term placental tissues; thereby confirming the limited utility of measuring average telomere lengths to assess changes in short telomeres. While our cohort was limited to the third trimester of pregnancy, our observation that even in the last weeks of human pregnancy, there is a significant increase in VST is consistent with the findings from Lai and colleagues [13]. One important difference between our current study and prior studies, including the one reported by Lai et al., is that the placental tissues in the prior studies were collected after the onset of labor [13, 19], making it difficult to determine if the process of undergoing labor had any effect on telomere lengths. By including only placentas from women who underwent Cesarean delivery prior to the onset of labor in our cohort, we were able to avoid this potentially confounding issue in our analysis.

Several obstetric factors have been shown to accelerate shortening of mean telomere length in human placenta and fetal cord blood, some of which include fetal growth restriction [20–22], maternal smoking [23], maternal air quality exposure [24], and fetal male sex [10]. Our study design did not allow us to evaluate the impact of these factors on VST between preterm and term placenta, because we did not include any growth restricted pregnancies and did not have maternal air quality data. Because of the small number of women in the other categories, our study was also not able to assess the effects of maternal smoking or fetal sex on the placental telomere lengths.

Our study has several strengths. First, this is a novel adaptation of telomere restriction fragment technique to assess the relative quantity of short telomeres in human tissue, previously demonstrated only in mouse tissue. Second, this study included a greater number of samples than most other published studies of telomere length in human placental tissue. Our study is one of the only telomere placental studies that exclusively examined pre-labor samples, allowing us to rule out labor itself as a confounder in regard to the observed telomere alterations.

The limitations of this study include a gestational age clustering of samples close to 36 weeks in the preterm group, as opposed to an even distribution across the third trimester. Our sample collection represents the demographics of our region of practice and may not be generalizable to large US or international populations. Our study did not include additional maternal or fetal tissue types leading to the inability to perform non-gestational tissue control analyses. Finally, our sample collection may have been susceptible to confounding by indication for delivery, which may have been incompletely captured by demographic and clinical data alone.

## Conclusions

In conclusion, VSTs are increased in term placentas compared to preterm placentas prior to the onset of labor, suggesting that placental telomeres progressively shorten as pregnancies approach term, i.e., 40 weeks of gestation. Further characterization of the biologic role of short telomeres in the placenta and their relationship with the timing for labor have the potential to

reveal fundamental insights about the biologic clock that determines the onset of parturition, along with the abnormal timing events leading to spontaneous preterm birth.

## Author Contributions

**Conceptualization:** Paula K. Edelson, Michala R. Sawyer, Kathryn J. Gray, David E. Cantonwine, Thomas F. McElrath, Mark Phillippe.

**Data curation:** Paula K. Edelson, Michala R. Sawyer, Kathryn J. Gray, David E. Cantonwine, Thomas F. McElrath, Mark Phillippe.

**Formal analysis:** Paula K. Edelson, Michala R. Sawyer, Mark Phillippe.

**Funding acquisition:** Mark Phillippe.

**Investigation:** Kathryn J. Gray, David E. Cantonwine, Mark Phillippe.

**Methodology:** Michala R. Sawyer, Kathryn J. Gray, David E. Cantonwine, Mark Phillippe.

**Project administration:** Michala R. Sawyer, David E. Cantonwine, Mark Phillippe.

**Resources:** Thomas F. McElrath.

**Supervision:** Mark Phillippe.

**Writing – original draft:** Paula K. Edelson.

**Writing – review & editing:** Paula K. Edelson, Kathryn J. Gray, David E. Cantonwine, Thomas F. McElrath, Mark Phillippe.

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
