## [Decision Letter · Decision Letter 0]

3 May 2022

PONE-D-22-06161Increase in short telomeres during the third trimester in human placentaPLOS ONE

Dear Dr. Edelson,

Thank you for submitting your manuscript to PLOS ONE. After careful consideration, we feel that it has merit but does not fully meet PLOS ONE’s publication criteria as it currently stands. Therefore, we invite you to submit a revised version of the manuscript that addresses the points raised during the review process. As you can see the referees find the analysis preliminary and require more data to warrant publication. They raise different concerns that I anticipate that you should be able to address. I would therefore like to invite you to submit a revised version.  For such a revision to be successful, it will however be important to fully resolve all of referee #1's concerns.

We look forward to receiving your revised manuscript.

Kind regards,

Khursheed Iqbal, Ph.D

Academic Editor

PLOS ONE

Journal Requirements:

(KJG is supported by NIH/NHLBI K08 HL146963. The funders had no role in study design, data collection and analysis, decision to publish, or preparation of the manuscript.)

MP is supported by Burroughs-Wellcome Fund. The funders had no role in study design, data collection and analysis, decision to publish, or preparation of the manuscript.

Reviewers' comments:

Reviewer's Responses to Questions

**Comments to the Author**

1. Is the manuscript technically sound, and do the data support the conclusions?

Reviewer #1: No

Reviewer #2: Partly

2. Has the statistical analysis been performed appropriately and rigorously? 

Reviewer #1: No

Reviewer #2: Yes

3. Have the authors made all data underlying the findings in their manuscript fully available?

Reviewer #1: Yes

Reviewer #2: Yes

4. Is the manuscript presented in an intelligible fashion and written in standard English?

Reviewer #1: Yes

Reviewer #2: Yes

5. Review Comments to the Author

Reviewer #1: Here, the authors assessed the ratio of short telomeres in placental tissue of patients who underwent Cesarean section in the third trimester. To quantify the short telomeres, the genomic DNA (gDNA) was isolated from placenta samples and digested by restriction enzymes (HinfI and RsaI) with a 4 and 5 base pair recognition sequence to release the telomeres. The digested gDNA was then size fractionated using a specific electrophoresis system and the fractions of 500 bp, 1 kb and 3 kb were PCR amplified with telomere-specific primers. The authors concluded `that placental telomeres progressively shorten as pregnancies approach 40 weeks of gestation`(ll225-226).

A weak point of this study is the strong experimental bias to detected short telomeres, without doing relevant controls. For example, only placental samples were analyzed, why didn´t the authors included chorioamniotic tissue, cord, cord blood, and blood from the mothers? The determination of mean telomere size would also improve the validity of the data. In the applied method the authors cannot discriminate between cell-free DNA and cellular DNA. Thus the first and last sentence in the Abstract - `An increase in telomere shortening in gestational tissues has been proposed as a mechanism for the initiation of parturition` (ll46-47); and `…thereby supporting the hypothesis that increasing short telomeres at term contributes to the mechanism leading to parturition`- are at least not rigorously tested. Sectioning and immunochemical staining of placental tissues was not done.

From the Materials and Methods it is unclear how the authors standardized the tissue sampling. The authors do not mention whether there were abnormal placentas, and if so, how many. With regard to the DNA fractionation, what do 500 bp, 1 kb and 3 kb mean?, e.g. is the 1 kb fraction exactly 1 kb, from > 500 bp to 1 kb, or 0 to 1 kb? Would sampling of a 0 to 3 kb fraction also yield an increased short telomere ratio, is there an experimental bias that produces significant increases in the 500 bp and 3 kb fractions? How did the authors confirm the completeness of DNA restriction, and the absence of star activity?

Apoptosis is a well described process in placenta maturation, and contribute to DNA fragmentation (also of telomeres). The here presented correlation between increased ratio of short telomeres with increased gestational age cannot elucidate whether this is a specific mechanism of telomer shortening or just an unspecific byproduct of apoptosis.

In conclusion, the presented data represent a preliminary draft and need supportive analyses.

Reviewer #2: In this manuscript, Edelson et al showed a significant increase in very short telomeres in human placental tissue at term. Authors should describe about the cell types of placenta in material and methods. The experimental content is relatively small, just a description of the phenomenon Its relevance and importance in parturition are too preliminary and speculative. However, I think is not ready for publication.

6. PLOS authors have the option to publish the peer review history of their article (what does this mean?). If published, this will include your full peer review and any attached files.

Reviewer #1: No

Reviewer #2: No

---

## [Author Response · Author response to Decision Letter 0]

16 Jun 2022

Journal Requirement Comments:

1. Manuscript style requirements: When submitting your revision, we need you to address these additional requirements.

Response: We have updated our formatting to reflect the style requirements as laid out in the manuscript formatting guidelines. 

Response: In our study, the need for consent was waived by the ethics committee (Institutional Review Board) because placental tissues are routinely discarded. The data used in subsequent analysis which involved abstraction of medical record were anonymized. We have updated the manuscript to clarify this point (Line 108-110). 

3. Thank you for stating in your Funding Statement: (KJG is supported by NIH/NHLBI K08 HL146963. The funders had no role in study design, data collection and analysis, decision to publish, or preparation of the manuscript.)

MP was supported by Burroughs-Wellcome Fund. The funders had no role in study design, data collection and analysis, decision to publish, or preparation of the manuscript.

Response: Thank you for bringing this to our attention. We have updated the funding statement as you have recommended above. This is included in the cover letter. 

Response: In our study, the need for consent was waived by the ethics committee (Institutional Review Board) because placental tissues are routinely discarded. The data used in subsequent analysis which involved abstraction of medical record were anonymized. We have updated the manuscript to clarify this point (Line 108-110).

Reviewer Comments to the Author:

Reviewer #1: 

A: “Here, the authors assessed the ratio of short telomeres in placental tissue of patients who underwent Cesarean section in the third trimester. To quantify the short telomeres, the genomic DNA (gDNA) was isolated from placenta samples and digested by restriction enzymes (HinfI and RsaI) with a 4 and 5 base pair recognition sequence to release the telomeres. The digested gDNA was then size fractionated using a specific electrophoresis system and the fractions of 500 bp, 1 kb and 3 kb were PCR amplified with telomere-specific primers. The authors concluded `that placental telomeres progressively shorten as pregnancies approach 40 weeks of gestation`(ll225-226). A weak point of this study is the strong experimental bias to detected short telomeres, without doing relevant controls. For example, only placental samples were analyzed, why didn´t the authors included chorioamniotic tissue, cord, cord blood, and blood from the mothers?” 

Response: The placentas were the tissue of interest collected for this study based on the previously published literature describing telomere shortening in the gestational tissues, especially the placenta. We have updated our limitations section to include the lack of control tissues (Line 225-226). 

B: “The determination of mean telomere size would also improve the validity of the data.” 

Response: The focus and novelty of this study was on short telomeres based on previously published reports demonstrating that the effects of telomere shortening are based on the increase in short telomeres, rather than any shift in the average telomere lengths (Hemann et al. Cell 2001;107:67-77). Also, there can occur a significant increase in short telomeres without any change in mean telomere lengths, so we disagree with the reviewer regarding value of having measured mean telomere lengths.

C: “In the applied method the authors cannot discriminate between cell-free DNA and cellular DNA.” 

Response: This statement is not correct. The telomeres were assayed after having extracted DNA from tissue specimens which contain genomic DNA, not cell free DNA. This method for DNA extraction and analysis has been previously published by our group (Phillippe et al. American Journal of Obstetrics and Gynecology 2019:220:496.e1-496.e8).

D: “Thus the first and last sentence in the Abstract - `An increase in telomere shortening in gestational tissues has been proposed as a mechanism for the initiation of parturition` (ll46-47); and `…thereby supporting the hypothesis that increasing short telomeres at term contributes to the mechanism leading to parturition`- are at least not rigorously tested.” 

Response: The reviewer needs to be aware of the large body of scientific evidence already supporting the premise that gestational length (and thus the timing for the onset of parturition) is based on a telomere gestational clock mechanism (ie Lai et al Sci Rep 2021:11:5115,Wilson et al Placenta 2016:48:26-33, Phillippe et al Reprod Sci 2015:22:1186-201, and many others). We acknowledge that this study does not definitively test the hypothesis that short telomeres control parturition, but it does provide novel and important data in support of this association. 

E: “Sectioning and immunochemical staining of placental tissues was not done.” 

Response: This is accurate; however, this comment appears irrelevant. These laboratory procedures were not part of the research engaged in our study.

F: “From the Materials and Methods it is unclear how the authors standardized the tissue sampling. The authors do not mention whether there were abnormal placentas, and if so, how many.” 

Response: As described in the Materials and Methods section, placentas were screened for eligibility criteria. Those placentas meeting inclusion criteria were collected and samples were taken from four representative 1-1.5 cm full thickness sections of the placental disk with a scalpel, and then rinsed in phosphate-buffered saline prior to being stored at -80� C. The manuscript has been updated to clarify this point (Line 111-112). 

The determination of “normal” vs “abnormal” placenta is more challenging, and is outside of the scope of our study. We have included extensive clinical data to correlate pregnancy co-morbid conditions such as hypertension, tobacco use, diabetes, etc., but our specimens did not undergo formal pathology examination by a pathologist to give a formal pathologic diagnosis of the placental tissue. 

G: “With regard to the DNA fractionation, what do 500 bp, 1 kb and 3 kb mean?, e.g. is the 1 kb fraction exactly 1 kb, from > 500 bp to 1 kb, or 0 to 1 kb? Would sampling of a 0 to 3 kb fraction also yield an increased short telomere ratio, is there an experimental bias that produces significant increases in the 500 bp and 3 kb fractions?”

Response: Size fractions were based on standard DNA gel electrophoresis techniques and size markers, so the 500 bp, 1 kb and 3 kb sizes are not exact, but narrow ranges DNA fractions at these sizes. Fraction sizes chosen for this study were arbitrary, but provided representative short telomere fractions compared to mean young adult telomeres in the 10 - 15 kb ranges. Obviously, if we had made our size collection ranges bigger (ie 0 - 500 bp as suggested by the reviewer), we would have measured more short telomeres using our qPCR technique.

H: “How did the authors confirm the completeness of DNA restriction, and the absence of star activity?” 

Response: The restriction digests were performed for the time periods and under the conditions recommended by the manufacturer, and also consistent with previous published reports using the classic telomere TRF (telomere restriction fragment); therefore, we validly assumed that digestions would be complete and absent abnormal enzyme STAR activities.

I: “Apoptosis is a well described process in placenta maturation, and contribute to DNA fragmentation (also of telomeres). The here presented correlation between increased ratio of short telomeres with increased gestational age cannot elucidate whether this is a specific mechanism of telomere shortening or just an unspecific byproduct of apoptosis.” 

Response: This reviewer is correct that there is a close relationship between short telomeres and apoptosis. Our premise is that the short telomeres lead to apoptosis, rather than the reverse. Also, during apoptosis, the DNA is fragmented into much smaller sizes consistent with cell free DNA found in the plasma (i.e. about 200-400 bp sizes). This might spuriously affect our 500 bp measurements, but not the larger 1 and 3 kb measurements.

J: “In conclusion, the presented data represent a preliminary draft and need supportive analyses.” 

Response: We respectfully disagree. Our studies are not definitive, but they are novel and provide the basis for ongoing similar research studies. As we state in our conclusion, the novel observations made in our study support the hypothesis that telomere shortening is associated with placental aging and maturation, but that certainly further characterization of the role of telomere length in influencing the onset of parturition is needed. 

Reviewer #2: 

A: “In this manuscript, Edelson et al showed a significant increase in very short telomeres in human placental tissue at term. Authors should describe about the cell types of placenta in material and methods.” 

Response: As is the case with all tissue biopsy studies for telomere lengths, the tissues are composed of two or more cell types, and the telomere measurements consist of a composite of the mix. The same is true for the placenta which is composed of trophoblast cells, stromal fibroblasts, fetal endothelial cells, immune cells, etc. 

B: “The experimental content is relatively small, just a description of the phenomenon. Its relevance and importance in parturition are too preliminary and speculative. However, I think is not ready for publication.” 

Response: As noted above, we respectfully disagree. Our studies are not definitive, but they are novel and provide the basis for ongoing study into this important topic.

---

## [Editor Report · Decision Letter 1]

30 Jun 2022

Increase in short telomeres during the third trimester in human placenta

PONE-D-22-06161R1

Dear Dr. Edelson,

We’re pleased to inform you that your manuscript has been judged scientifically suitable for publication and will be formally accepted for publication once it meets all outstanding technical requirements.

Kind regards,

Khursheed Iqbal, Ph.D

Academic Editor

PLOS ONE
---

## [Editor Report · Acceptance letter]

5 Jul 2022

PONE-D-22-06161R1 

Increase in short telomeres during the third trimester in human placenta 

Dear Dr. Edelson:

I'm pleased to inform you that your manuscript has been deemed suitable for publication in PLOS ONE. Congratulations! Your manuscript is now with our production department. 

Kind regards, 

on behalf of

Dr. Khursheed Iqbal 

Academic Editor

PLOS ONE